# OpenReview forum: "Towards Robust Agentic Systems through Generative Flow Exploration of Primitives"
_ICLR.cc/2026/Conference — Submitted to ICLR 2026_

### Official Review · Reviewer_5XBL · 2025-10-30

**Soundness:** 2
**Presentation:** 2
**Contribution:** 2
**Rating:** 2
**Confidence:** 3

**Summary:**

This paper proposes a framework called AutoRAS for automatically designing robust agentic systems. The main idea is to represent the system design as a sequence of "primitives" and then use a flow-based optimization method to find good sequences. It also uses "safety signals" from running the system to feed back into the design process. Evaluation on several benchmarks (MMLU, MATH, etc.) under different attacks shows that this method produces systems that are more robust than baselines.

**Strengths:**

I agree that multi-agent systems need to be more robust, and investigating how to automatically design them is an important issue. This paper focuses on this specific problem, proposes a method for doing it, and shows that it works better than other methods. The findings align with expectations, and the idea of using primitives and flow exploration is interesting for designing agent systems.

**Weaknesses:**

First, I find the proposed method overly complex. It combines a sequence generator, GFlowNets, Trajectory Balance, and a separate "analyzer" LLM for "textual gradients". It feels like a lot of moving parts just to generate a simple workflow. The 'primitives' idea seems like a re-branding of what many people are already doing with graph generation or even just scripting agent interactions. I find it hard to tell how this is fundamentally different from other automated agent design work like GPTSwarm or AFlow, which are cited but the distinction isn't made clear enough.

Second, the whole contribution seems to hinge on the "primitives". But the authors just hand-crafted these primitives themselves in Appendix D . The system is just learning how to stack these pre-defined blocks (like SAFE_Filter or AGT_COT). This doesn't seem to solve the design problem; it just pushes it down a level. The system isn't inventing a new robust workflow, it's just picking from a list of safety tools the authors already gave it. I would argue this is a much simpler search problem than the paper claims, and it's not clear if this approach can generalize beyond the authors' hand-picked primitive set.

Third, I have concerns about the evaluation. The paper's method, AutoRAS, is explicitly trained using a reward function that includes robustness scores (r_ext, r_int). It's no surprise that it does well on robustness when it's the only method being optimized for it. I find it hard to tell if this is a fair comparison against baselines like AFlow or GPTSwarm, which were likely designed for performance, not for these specific attacks. Furthermore, the dataset sizes in Table 4 are extremely small. For ProgramDev, it only uses 6 training samples, and for MSMARCO, only 20. How can a complex GFlowNet model learn anything from 6 or 20 samples? This makes me question if the system is really 'learning' a general design policy or just overfitting to a few examples.

Some typos:  line 155, the figure index is missing.

**Questions:**

See weakness.

---

> ### Author Response · Authors · 2025-11-24
> **RESPONSE to Reviewer 5XBL [PART 1/3]**
>
> We would like to express our sincere respect for your insightful review!  In response to your comments, we have carefully prepared a point-by-point reply:
>
> ---
> >**Weakness 1**. First, I find the proposed method overly complex. It combines a sequence generator, GFlowNets, Trajectory Balance, and a separate "analyzer" LLM for "textual gradients". It feels like a lot of moving parts just to generate a simple workflow. The 'primitives' idea seems like a re-branding of what many people are already doing with graph generation or even just scripting agent interactions. I find it hard to tell how this is fundamentally different from other automated agent design work like GPTSwarm or AFlow, which are cited but the distinction isn't made clear enough.
>
> **1.Regarding the concern about methodological complexity.**
>
> - **The design is conceptually simple and follows a clear, modular logic, although it contains several components.** The system first represents an agentic system through a primitive sequence, then optimizes this representation using a lightweight policy model with GFlowNets and a single-step Trajectory Balance objective, and finally applies a textual-gradient step to refine prompts.
>
> - **In practice, AutoRAS is computationally light.** The policy model has only **4.29M parameters (49.22 MB)**, training uses about **450 MB of GPU memory,** and a full run takes **2–6 hours**, with over **90%** of the time spent waiting for LLM responses rather than running optimization. Furthermore, as demonstrated in **Appendix H.6**, the training exhibits rapid and stable convergence, typically stabilizing after processing approximately 30 training trajectories.
> - **This structure is not unusually complex compared to existing automated agentic system design frameworks.**  Prior work such as GPTSwarm, MaAS, and AFlow follows the same two-stage paradigm: (1) defining a representation of an agentic system, and (2) optimizing that representation with a suitable learning objective.  Our choice of primitives and GFlowNets with TB loss is a instantiation of this paradigm.
>
> **2.Key difference between existing method.**
>
> - **Representation:** Primitive sequences capture richer structure and behavior. AutoRAS represents an agentic system as a primitive sequence that jointly encodes structural and behavioral patterns.
>
> - **Optimization:** GFlowNets + Trajectory Balance provide stable and efficient structure search. We use GFlowNets with TB loss, which avoids long-horizon credit assignment, offers higher stability and naturally handles multiple high-reward designs (equifinality) in large discrete spaces.
>
> - **Robustness:** AutoRAS integrates robustness signals throughout the design phase and execute phase.
>
> **Table 1. Comparison of representation methods**
>
> | Representation | Strengths | Limitations |
> |---|---|---|
> | Neural Network | Captures complex behavioral dependencies | Implicit structure and control flow|
> | Graph | Clearly expresses workflow topology | Lacks behavioral semantics (e.g. reasoning mode or control conditions) |
> | Code | Highly expressive; precise control flow and arbitrary interaction logic | Fragile and difficult to constrain  |
> | Operator| Clear behavioral semantics; easy to compose  | No explicit structural semantics; Requires operator design|
> | Primitive (Ours) | Unifies structural and behavioral semantics; compositional and searchable | Requires vocabulary design |
>
> **Table 2. System-level capability comparison**
>
> | System  | Representation | Topology search | Behavioral variation | Robustness design | Optimization  | Prompt Refine |
> |---|---|---|---|---|---|---|
> | Dylan | Neural Network | × | ✓ | ×| LLM+Rule | × |
> | GPTSwarm | Graph | ✓|× | ✓ | Edge Optimization+Policy Gradient | ✓ |
> | ADAS  | Code | ✓  | ✓  | × | LLM  | ✓ |
> | AFlow | Operator| ✓ | ✓| × | LLM+MCTS | ✓ |
> | AgentPrune | Graph|✓| × |✓| Graph Sparsification+Policy Gradient | × |
> | G-designer | Graph  | ✓  | × |✓| GCN+Policy Gradient | × |
> | MaAS  | Operator| ×| ✓| × | Agentic Supernet+Policy Gradient| ✓|
> | AutoRAS| Primitive | ✓| ✓|✓| GFlowNet + TB loss |✓|
>
> ---

---

> ### Author Response · Authors · 2025-11-24
> **RESPONSE to Reviewer 5XBL [PART 2/3]**
>
> > **Weakness 2**. Second, the whole contribution seems to hinge on the "primitives". But the authors just hand-crafted these primitives themselves in Appendix D . The system is just learning how to stack these pre-defined blocks (like SAFE_Filter or AGT_COT). This doesn't seem to solve the design problem; it just pushes it down a level. The system isn't inventing a new robust workflow, it's just picking from a list of safety tools the authors already gave it. I would argue this is a much simpler search problem than the paper claims, and it's not clear if this approach can generalize beyond the authors' hand-picked primitive set.
>
> **1. Clarifying the design problem.**
>
> Automated agentic-system design, as defined in MaAS (ICML 2025 Oral) and AFlow (ICLR 2025 Oral), is not about inventing new atomic abilities. It is about discovering how existing agent behaviors should be composed, coordinated, and sequenced to produce more effective systems. Primitives serve as abstractions of these behaviors and interaction patterns—not as the solution itself. Thus, AutoRAS follows the standard formulation of the problem.
>
> **2. Clarifying robustness.**
>
> Our work does not aim to create new safety tools; instead, we address the harder problem of when, where, and how to use these tools within an agentic system to maximize robustness without excessive cost. This trade-off between robustness and efficiency is the core difficulty we optimize, not the tools themselves.
>
> **3. The search problem is not simple.**
>
> While primitive sequence generation can be seen as a search problem, it operates over a large combinatorial space combining structure, behavior, and safety. With L = 16, the space already contains approximately $10^6$~$10^8$ valid workflows under stack-based legality constraints. This makes the optimization substantially more challenging than “stacking blocks.” GFlowNets with Trajectory Balance provide single-step training, stability, and support for equifinality, making them more suitable than MCTS-based methods (AFlow) or policy gradient approaches (GPTSwarm) for this large discrete space.
>
> **4. Generalization of primitives.**
>
> First, our goal is not to enumerate all possible agentic systems, since many different workflows can achieve similar outcomes (equifinality). Instead, we seek an expressive yet manageable representation space. The primitive vocabulary is **extensible**, and new primitives can be added when needed. What matters is providing a sufficiently rich abstraction so that effective workflows can emerge within the space. Futhermore, we conduct a primitive sensitivity experiments in Table 3 and 4. The results show that:
>
> - Removing primitives (especially functional or safety ones) significantly harms performance.
> - Adding more primitives yields only marginal gains.
> - Behavioral primitives influence performance the most;
> - Safety primitives influence both robustness and accuracy;
> - Structural primitives have the smallest effect;
> - The smallest primitive sets perform the worst.
>
> These trends match expectations and confirm that the current primitive set offers a reasonable and robust expressiveness level rather than relying on completeness.
>
> **Table 3. Settings for Primitive Sensitivity Experiments**
>
> | Name | Description |
> | ---| ---|
> | Vocab 1: Minimal Behavior | Keep only two primitives (**AGT_DIRECT**, **AGT_ENS**) in Behavioral Primitives |
> | Vocab 2: Minimal Safety | Remove all safety primitives. |
> | Vocab 3: Minimal Function | Keep only **AGT_DIRECT** among functional primitives. |
> | Vocab 4: Minimal Structure | Keep only **BEG\SEP\CTRL_SEQ** among structural primitives.  |
> | Vocab 5: Original Primitive Set (Ours) | The complete primitive repository used in the main paper. |
> | Vocab 6: Add REACT |Adding **AGT_REACT** (new functional behavioral).|
> | Vocab 7: Add REACT&CYCLE  | Adding **AGT_REACT** and **CTRL_CYCLE** (new structural primitive). |
>
> **Tabel 4. Primitive Vocabulary Sensitivity Analysis.** Num denotes the count of primitives in the vocabulary. Costs are calculated based on API token usage.
>
> | Name| num | Prompt token | Completion token | cost | length | vanilla | attack |
> | ---|---| ---| ---| ---|---|---| ---|
> | Vocab 1 | 9  | 1,831,814 | 172,449 | 0.4239 | 6.77| 77.78 | 73.20|
> | Vocab 2 | 12| 2,416,341 | 482,617| 0.6520 | 11.79  | 81.70| 75.16|
> | Vocab 3 | 13| 1,861,450 | 454,542 | 0.5519 | 13.56 | 81.05| 79.74|
> | Vocab 4 | 13| 1,765,934 | 534,052 | 0.5853 | 15.00 | 81.70 | 78.43|
> | Vocab 5 | 17| 2,007,650 | 607,423 | 0.6655 | 13.70 | 83.01 | 82.35|
> | Vocab 6 | 18| 2,185,156| 655,002  | 0.7208 | 13.86 | 83.66 | 81.70|
> | Vocab 7 | 19| 2,859,977| 1,099,952 | 1.0890 | 13.01| 83.01 | 82.35|
>
> ---

---

> ### Author Response · Authors · 2025-11-24
> **RESPONSE to Reviewer 5XBL [PART 3/3]**
>
> > **Weakness 3**. Third, I have concerns about the evaluation. The paper's method, AutoRAS, is explicitly trained using a reward function that includes robustness scores. It's no surprise that it does well on robustness when it's the only method being optimized for it. I find it hard to tell if this is a fair comparison against baselines like AFlow or GPTSwarm, which were likely designed for performance, not for these specific attacks. Furthermore, the dataset sizes in Table 4 are extremely small. For ProgramDev, it only uses 6 training samples, and for MSMARCO, only 20. How can a complex GFlowNet model learn anything from 6 or 20 samples? This makes me question if the system is really 'learning' a general design policy or just overfitting to a few examples.
>
> **1.AutoRAS is not the only agentic-system design framework that considers robustness.**
>
> Several baselines we compare with, including GPTSwarm, AgentPrune, and G-designer, explicitly discuss robustness in their paper. Moreover, in automated agentic system design, the objective is to learn how agent behaviors and interactions should be composed to achieve specific design goals, including robustness. Using robustness-related signals in the reward is therefore expected and appropriate for this problem setting.  AutoRAS differs mainly in that it integrates robustness dynamically throughout both design and execution, adopting proactive defensive strategies instead of relying on fixed heuristics that react only after an attack.
>
> **2.The size of the training sets.**
>
> - **We follow the dataset split protocol used in AFlow (ICLR 2025 Oral) and MaAS (ICML 2025 Oral), using a 1:4 ratio between training and test queries.** To further examine the effect of limited training data, we evaluate different numbers of training queries N on two relatively small datasets, **MSMARCO** and **ProgramDev**, as reported in Tables 5 and 6. The results show that AutoRAS can effectively learn to generate strong and generalizable sequences with only a small number of training samples.
>
> - **Moreover, each training query is sampled K times** (with K = 4 as the default, detailed in **Appendix H.1**). So the effective number of training trajectories is $N\times K$, providing sufficient signal for learning. Since the ProgramDev dataset contains only 6 training queries after the 1:4 split, we further evaluated different values of K on ProgramDev in **Tabel 7**. The results show that, when the training set is extremely small, increasing the sampling count K can indeed yield additional gains, illustrating the potential of our approach under low-data conditions. Nonetheless, we still recommend K = 4 as the default, as it provides a good balance between performance and cost across datasets.
>
> - **In addition, AutoRAS learns a design policy.** The search space consists of workflow structures, and the policy model is lightweight (only 4.29M parameters), so it does not require large datasets. The reward signals are task-agnostic and provide strong supervision, enabling GFlowNets to efficiently explore the structure space even with limited data. This is further supported by the Training Query Sensitivity results (Table 5), which demonstrate stable performance across different amounts of training data.
>
> **Tabel 5. Training queries(N) sensitivity analysis on MSMARCO**.
>
> |Number of training queries |5|10|15|20|
> |---|---|---|---|---|
> |vanilla | 86.25 | 87.50 | 90.00 | 90.00 |
> |attack | 83.75 | 83.75 | 88.75 | 88.75 |
>
> **Tabel 6. Training queries(N)sensitivity analysis on ProgramDev**.
>
> |Number of training queries | 2 | 4 | 6 |
> |---| ---|---|---|
> |vanilla | 52.08 | 60.41 | 66.67 |
> |attack  | 50.00 | 56.25 | 62.50 |
>
> **Tabel 7. Sampling times(K) sensitivity analysis on ProgramDev**.
>
> | K| 2 | 4 | 6 | 8 |
> |---|---|---|---|---|
> |vanilla|56.25 | 66.67 | 66.67 | 70.83 |
> |attack| 52.08 | 62.50 | 64.58 | 66.67 |
>
> **3.Transfer experiments.**
>
> We also evaluate cross-dataset transferability in **Table 8**. The result shows that a policy trained on one dataset transfers well to different datasets. The results demonstrates that AutoRAS learns generalizable structural design principles rather than memorizing a small number of instances.
>
> **Tabel 8. Datasets transferability analysis.** Rows denote the dataset used for training; columns denote the dataset used for testing.
>
> |  | Test on MMLU |  | Test on MATH ||
> |---|---|---|---|---|
> | **Training Set** |Vanilla| Attack | Vanilla | Attack |
> | MMLU |83.01|82.35|56.38|55.56|
> | MATH |83.66|81.70|57.41|54.94|
> | MSMARCO |81.70|81.05|56.58 | 55.97|
> | programdev|83.01 |80.39|56.79| 56.38|
>
> Together, these results indicate that AutoRAS is not overfitting to a tiny dataset. It learns transferable and dataset-agnostic workflow patterns that provide robustness improvements.
>
> ---
> > Typos.
>
> We have corrected the typos in the revised version accordingly.
>
> ---
> We sincerely hope that the clarifications and additional experiments provided above address your concerns!

---

### Official Review · Reviewer_2i3R · 2025-11-01

**Soundness:** 4
**Presentation:** 3
**Contribution:** 3
**Rating:** 6
**Confidence:** 4

**Summary:**

This paper introduces AutoRAS, a framework for the Automated design of Robust Agentic Systems. It formulates agentic system design as a sequence generation problem over symbolic primitives that encode both structural and behavioral elements. The framework integrates flow-based optimization (GFlowNet) and textual feedback from execution traces to iteratively refine system robustness and reliability.

**Strengths:**

1. The method reframes agentic system construction as primitive-sequence generation with built-in structural validation and dynamic safety signals, offering a novel and generalizable paradigm.

**Weaknesses:**

1. While performance gains are strong, the paper does not detail the training overhead or inference efficiency compared to simpler baselines.
2. Could the abstraction limit the flexibility of agentic systems? For example, the real-world agents may have more diverse multi-agent systems and orchestration.

**Questions:**

1. How sensitive is AutoRAS to the design of the primitive vocabulary—does adding or removing primitives materially affect convergence and robustness?
2. Could the authors provide quantitative measures of computational efficiency and cost, particularly during the flow exploration phase?

---

> ### Author Response · Authors · 2025-11-23
> **RESPONSE to Reviewer 2i3R PART [1/2]**
>
> We thank the reviewer for the clear and helpful comments. We provide below a point-by-point response to each issue raised.
>
> ---
>
> > **Weakness 1 and Question 2.** While performance gains are strong, the paper does not detail the training overhead or inference efficiency compared to simpler baselines, and it would be helpful to see quantitative measures of computational efficiency and cost.
>
> **1.Cost analysis.**
>
> We provide a detailed cost comparison on MMLU against 5 baselines in Table 1. AutoRAS is **slightly more costly but highly effective**. Because AutoRAS performs more flexible system exploration and incorporates additional safety considerations during execution, yet it remains the **second-lowest** in total cost among all methods. Most importantly, the performance and robustness far outweigh this overhead. AutoRAS achieves 83.01% → 82.35% under attack, whereas other methods suffer substantially larger drops (−14% to −27%). This demonstrates that the added structural search yields significant robustness benefits relative to its cost, and AutoRAS already attains the strongest vanilla performance among all baselines.
>
> **Table 1. Cost analysis across baselines.** GPTSwarm, AgentPrune, and G-designer explicitly discuss robustness in their papers, while AFlow (ICLR 2025) and MaAS (ICML 2025) represent recent automated agentic system design methods.
>
> | | Training| | | Inference | | | Total | Performance | |
> |---|---|---|---|---|---|---|---|---|---|
> | **Method** | Prompt token | Completion token | Cost($) | Prompt token | Completion token | Cost($) | Cost($) |vanilla| attack |
> |GPTSwarm| 3,594,420| 1,065,580|1.18| 1,124,913| 430,268|0.43| 1.61 | 75.82|71.24|
> | Agentprune | 844,814| 191,800 | 0.24|3,780,913| 861,543| 1.08  | 1.33  | 81.70| 76.47  |
> | AFlow | 10,117,493   | 1,153,666 | 2.21| 1,033,808 | 161,119  | 0.25    | 2.46 | 82.35| 70.58  |
> | G-designer | 598,010 | 115,704 |0.16| 2,840,066 | 528,981   | 0.74  | 0.90  | 82.35 | 73.86  |
> | MaAS |392,428| 306,049 |0.24|170,956|107,215|0.09|0.33| 81.17 |66.01|
> | ours |1,029,399| 299,883 | 0.33  | 978,251| 307,540| 0.33| 0.67  | 83.01| 82.35  |
>
> **2.Computational efficiency.**
>
>  Beyond the LLM calls, **the computational footprint of AutoRAS is very small**. Our policy model has only 4.29M parameters and occupies 49.22 MB on disk, requiring roughly 450 MB of GPU memory during training. A full training run takes 2–6 hours, and more than 90% of the time is spent waiting for backbone LLM responses. This shows that the flow-exploration phase adds only a modest overhead and is not the dominant cost.
>
> ---
> >**Weakness 2.** Could the abstraction limit the flexibility of agentic systems? For example, the real-world agents may have more diverse multi-agent systems and orchestration.
>
> 1. **AutoRAS focuses on discovering strong workflows within a sufficiently expressive primitive space, not to cover every possible design**. This follows the standard goal in automated agentic system design.  No existing representation, including graph, node, or operator, can theoretically cover all design patterns.  Our primitive sequence formulation already spans a large space: with L = 16, it can express roughly $10^6$~$10^8$distinct workflows by jointly modeling both structural and behavioral patterns.
> 2. **GFlowNets naturally support equifinality.** Multiple agentic systems may achieve similar reward, allowing AutoRAS to find alternative patterns even when certain primitives are absent. For example, loop-like behaviors can be represented through unfolded sequential structures without requiring an explicit loop primitive. Our primitive sensitivity analysis further confirm this, as detailed in our response to Q1 in the second part.
> 3. In Tabel 2, we compare the existing representation methods for agentic system.Primitive provides a more flexible and expressive representation compared to prior representations due to the combination of structural and behavioral primitives.
>
> **Tabel 2. Comparison between primitives and other representation methods.**
>
> | Representation | Strengths | Limitations |
> |---|---|---|
> | Neural Network   | Captures complex behavioral dependencies| Implicit structure and control flow|
> | Graph | Clearly expresses workflow topology | Lacks behavioral semantics (e.g. reasoning mode or control conditions) |
> | Code | Highly expressive; precise control flow and arbitrary interaction logic | Fragile and difficult to constrain  |
> | Operator | Clear behavioral semantics; easy to compose | No explicit structural semantics; Requires operator design|
> | Primitive (Ours) | Unifies structural and behavioral semantics; compositional and searchable | Requires vocabulary design |
>
> ---

---

> ### Author Response · Authors · 2025-11-23
> **RESPONSE to Reviewer 2i3R [PART 2/2]**
>
> > **Question 1.** How sensitive is AutoRAS to the design of the primitive vocabulary—does adding or removing primitives materially affect convergence and robustness?
>
> **We evaluate the sensitivity of the primitive vocabulary on the MMLU dataset.** Table 3 lists the configurations we tested, and Table 4 reports the resulting performance. Our analysis varies both the number and diversity of primitives and measures their impact on performance, robustness and cost. The results clearly show that the current primitive design is well-balanced:
>
> - Removing primitives substantially degrades performance, especially when functional or safety primitives are removed.
> - Adding additional primitives produces only marginal gains, indicating diminishing returns beyond the current set.
> - Structural primitives have the smallest effect on accuracy, while behavioral primitives have the strongest, and safety primitives influence both robustness and accuracy.
> - The smallest primitive sets perform the worst, confirming that the existing design represents a good middle ground between expressiveness and efficiency.
>
> The results confirm that the primitive set is extensible, and the impact of different primitives matches expectations, indicating that the current design is reasonable.
>
> **Table 3. Settings for Primitive Sensitivity Experiments**
>
> | Name                         | Description                                                  |
> | ---------------------------- | ------------------------------------------------------------ |
> | Vocab 1: Minimal Behavior    | Keep only two primitives (**AGT_DIRECT**, **AGT_ENS**) in Behavioral Primitives |
> | Vocab 2: Minimal Safety      | Remove all safety primitives.                                |
> | Vocab 3: Minimal Function    | Keep only **AGT_DIRECT** among functional primitives.        |
> | Vocab 4: Minimal Structure   | Keep only **BEG\SEP\CTRL_SEQ** among structural primitives.  |
> | Vocab 5: Original Set (Ours) | The complete primitive repository used in the main paper.    |
> | Vocab 6: Add REACT           | Adding **AGT_REACT** (new functional behavioral).            |
> | Vocab 7: Add REACT&CYCLE     | Adding **AGT_REACT** and **CTRL_CYCLE** (new structural primitive). |
>
> **Tabel 4. Primitive Vocabulary Sensitivity Analysis.**
>
> |         | num  | Prompt token | Completion token | cost   | length | vanilla | attack |
> | ------- | ---- | ------------ | ---------------- | ------ | ------ | ------- | ------ |
> | Vocab 1 | 9    | 1,831,814    | 172,449          | 0.4239 | 6.77   | 77.78   | 73.20  |
> | Vocab 2 | 12   | 2,416,341    | 482,617          | 0.6520 | 11.79  | 81.70   | 75.16  |
> | Vocab 3 | 13   | 1,861,450    | 454,542          | 0.5519 | 13.56  | 81.05   | 79.74  |
> | Vocab 4 | 13   | 1,765,934    | 534,052          | 0.5853 | 15.00  | 81.70   | 78.43  |
> | Vocab 5 | 17   | 2,007,650    | 607,423          | 0.6655 | 13.70  | 83.01   | 82.35  |
> | Vocab 6 | 18   | 2,185,156    | 655,002          | 0.7208 | 13.86  | 83.66   | 81.70  |
> | Vocab 7 | 19   | 2,859,977    | 1,099,952        | 1.0890 | 13.01  | 83.01   | 82.35  |
>
> ---

---

### Official Review · Reviewer_DBFH · 2025-11-02

**Soundness:** 2
**Presentation:** 3
**Contribution:** 2
**Rating:** 4
**Confidence:** 4

**Summary:**

The paper introduces, AutoRAS, a framework for the Automated design of Robust Agentic Systems.
AutoRAS represents agentic systems as a sequence generation problem over symbolic primitives encoding both structural connections and behavioral actions. AutoRAS can be used to improve any agentic's system reliability. The authors also report experiment results on four datasets against 11 baselines showing how AutoRAS achieves the best result under attack.

**Strengths:**

- Significance: Designing robust agentic systems automatically is an important problem given the push in both academia and industry to move towards AGI. The majority of production agentic systems today need constant maintenance as the frontier models get updated or the tool APIs improve. Significant engineering time is spent on optimiznig the prompt and system design of agentic systems to adopt to the new model or tool api updates. Therefore, any system that can automate agent design reliably significantly contributes to all frontier agents in industry and academia.
- Clarity: The paper does a great job clarifying why its hard to design a reliable agent system automatically. It categorizes the challenges into three categories of entanglement, unpredictability, and equifinality and explains each category clearly.
- Novelty: The authors leverage two techniques that are useful. 1) they integrate robustness in the reward function. 2) they leverage textual gradients to refine the prompts.

**Weaknesses:**

- Application: Many of the real agentic systems working in production today leverage complex elements that cannot be mapped to the set of primitives the paper defines to begin with. For example, a coding agent such as cursor leverages fine-tuned models to apply a code change efficiently and reliably, which primitive can capture such modules? and how an automated system will arrive at such design? Many of the components in the current production agents come from carefully assessing the performance of different part of the system and crafting solutions to improve the performance. A system such as AutoRAS cannot replicate this process.
- Experiments: The manually crafted agentic systems that the authors have picked for evaluations are very primitive. It would be great if they could also compare with a production grade agentic system.

**Questions:**

- Why did the authors used smaller models the mini series from openAI and haiku and flash series from Anthropic and Google? Would the vanilla baseline results change switching to more powerful frontier models?

---

> ### Author Response · Authors · 2025-11-24
> **RESPONSE to Reviewer DBFH**
>
> We sincerely appreciate the reviewer’s insightful feedback. Below we provide a detailed point-by-point response addressing all concerns.
>
> > **Weakness 1.** Application: Whether AutoRAS can represent or design real production-grade agentic systems, since such systems (e.g., Cursor) rely on complex fine-tuned modules and hand-engineered pipelines that do not directly map to primitive set. Whether an automated method can reproduce the human engineering process behind these components.
>
> 1.	**Compatibility rather than replacement.** Production systems like Cursor indeed use internal multi-agent pipelines built from fine-tuned coding models, diff engines, and static analyzers. AutoRAS does not attempt to reimplement or approximate these highly specialized components. Such modules can simply be exposed as callable operators and encapsulated as primitives (e.g., as a low-level instance of our AGT_PROGRAM operator). AutoRAS can therefore leverage these capabilities directly rather than substitute them.
>
>  2.	**The scope of automated agentic system design.** Following prior work such as MaAS (ICML2025 Oral), AFlow (ICLR2025 Oral), and GPTSwarm (ICML2024 Oral), the goal of this line of research is not to design or train domain-specialized modules. Instead, the shared objective is to automatically explore and optimize the workflow-level structure of agentic systems: how modules are assembled, coordinated, sequenced, and safeguarded. Automated agentic system design therefore serves as a complement to hand-engineered production components rather than a replacement for them.
>  3.	**Methodological advantages of AutoRAS within this scope.** Under the same design objective, AutoRAS provides a flexible and efficient formulation by representing workflows as primitive sequences and optimizing them using GFlowNets. This enables systematic exploration of diverse compositions, effective handling of equifinality, and the integration of robustness signals into the design loop.
>  4.	**Extensibility of the primitive space.** The primitive set is not fixed. Any production component can be added as new primitives. AutoRAS operates on this extensible action space and does not assume a closed or simplified set of modules. We further discuss the extensibility of primitives and provide supporting experiments in **Appendix D.3**.
>
> ---
>
> > **Weakness 2.** Experiments: The manually crafted agentic systems that the authors have picked for evaluations are very primitive. It would be great if they could also compare with a production grade agentic system.
>
> **We have added comparisons with 3 production-grade agentic systems: Megenic-One [1], CAMEL [2], and OWL [3].**
> This experiment evaluates both vanilla performance, attack robustness, and the cost, following the same settings as in our main paper. Results show that AutoRAS maintains strong overall performance and consistently exhibits the smallest drop under attack, when compared with these production systems. Its higher cost arises from sampling several candidate workflows during training, which is intrinsic to automated system design.
>
> **Tabel 1. Comparison with production-grade agentic systems on MMLU and MATH.**
>
> ||MMLU||||Math||||
> | ---| --- | ---| ---| --- |---|---|---|---|
> |**System**|Vanilla|cost($)|Attack|cost($)|Vanilla|cost($)| Attack | cost($) |
> | megenic-one | 81.04| 0.27| 68.62  | 0.38| 45.88| 1.21| 23.25 | 1.62|
> | camel  | 69.93| 0.23 | 62.75  | 0.52| 46.70| 2.04| 41.98| 2.45 |
> | OWL   | 78.43| 0.65| 68.62  | 0.67| 50.41 | 3.23 | 45.68 | 3.31 |
> | AutoRAS(ours) | 83.01| 0.66 | 82.35| 0.67 | 57.41| 3.51 | 54.94| 3.48 |
>
> [1]Magentic-One: A Generalist Multi-Agent System for Solving Complex Tasks,Microsoft
>
> [2]CAMEL:Communicative Agents for “Mind”  Exploration of Large Language Model Society,NeurIPS 2023
>
> [3]OWL: Optimized Workforce Learning for General Multi-Agent Assistance in Real-World Task Automation,NeurIPS 2025
>
> ---
>
> >  **Question 1.** Why did the authors used smaller models the mini series from openAI and haiku and flash series from Anthropic and Google? Would the vanilla baseline results change switching to more powerful frontier models?
>
> **We used GPT-4o-mini as the main backbone due to practical constraints on API cost and throughput, following prior automated design work such as MaAS (ICML 2025 Oral) and AFlow (ICLR 2025 Oral).** To ensure that the observed gains are not tied to a specific backbone, we further evaluated AutoRAS on 3 additional models（DeepSeek-V3.1, Claude-3.5-Haiku, and Gemini-2.0-Flash), including the more capable DeepSeek-V3.1. Across all 4 backbones, AutoRAS consistently improves over each model’s own vanilla performance (in **Sec 5.3**), showing that the gains are independent of backbone strength.
>
> ---
>
> We sincerely hope that the clarifications and additional experiments provided above address your concerns!

---

### Official Review · Reviewer_zZPz · 2025-11-03

**Soundness:** 2
**Presentation:** 2
**Contribution:** 2
**Rating:** 4
**Confidence:** 3

**Summary:**

This paper introduces AutoRAS, a framework to automatically design robust, multi-agent LLM systems that are resilient to both external attacks and internal failures. The core idea is to represent the complex design of an agentic system—including its structure and behaviors—as a sequence generation problem using a defined set of symbolic "primitives." AutoRAS then uses flow-based optimization, guided by a novel dual-feedback loop of numeric rewards (e.g., accuracy) and textual safety signals (distilled from execution traces), to explore this design space. This method discovers agentic workflows that achieve state-of-the-art performance while exhibiting the smallest performance degradation under various adversarial attacks compared to existing methods.

**Strengths:**

- The paper reframes the agentic system design into a concrete sequence generation problem over symbolic "primitives." This abstraction makes the vast search space of possible designs (combining different structures, communication patterns, and agent behaviors) tractable and optimizable.
- AutoRAS embeds robustness directly into the optimization loop. The feedback signals about failure modes, allows the system to learn why designs fail and proactively favor more resilient architectures.
- The framework demonstrates empirical advantages with state-of-the-art accuracy on several benchmarks and shows the smallest average performance drop under four different attack types.
- The workflows demonstrate transferability and high performance when executed with different backbone models (e.g., GPT-40-MINI, DeepSeek-V3.1, Claude-3.5-Haiku)

**Weaknesses:**

- Potentially Saturated Benchmarks: The paper relies on benchmarks like MMLU and MATH, which are becoming saturated in the sense that top-tier SOTA models (like the latest GPT, Claude, or Gemini series) can already achieve very high performance. This makes it difficult to assess how much of the performance is from the novel AutoRAS framework versus the underlying power of the base LLM it uses.
- Complexity of Optimization: The system uses flow-based optimization to search the sequence space, which can be computationally expensive and complex to train. The performance is also sensitive to parameters like sequence length and training samples per iteration.
- Missing Cost-Benefit Analysis: The AutoRAS algorithm is significantly more complex than a standard single-agent baseline. The paper does not provide a direct comparison of the computational cost against these simpler baselines, making it hard to evaluate if the performance and robustness gains justify the added complexity.

- Dependency on the Primitive Set: The entire framework's success is contingent on the quality and completeness of the predefined "primitive" vocabulary. If a highly effective or robust design pattern is not expressible using the existing primitives, AutoRAS will be unable to discover it.
- Dependency on LLMs: The framework's core optimization loop is critically dependent on the quality of its own internal LLMs. The "monitor" (Sec 4.2) uses an LLM to detect failures and generate safety signals; if this LLM fails to detect a novel or stealthy attack, the system cannot learn to defend against it. Similarly, the "analyzer" (Sec 4.1) and "textual gradient" (Sec 4.3) rely on an LLM to refine prompts, meaning the quality of the final agent behaviors is capped by this meta-LLM's capabilities.

**Questions:**

There are a number of typos such as line 047 and line 155.

---

> ### Author Response · Authors · 2025-11-24
> **RESPONSE to Reviewer zZPz PART [1/3]**
>
> We would like to express our sincere appreciation for your thoughtful and constructive review. In response to your comments, we have carefully prepared a point-by-point reply as follows.
>
> > **Weakness 1:** otentially Saturated Benchmarks: The paper relies on benchmarks like MMLU and MATH, which are becoming saturated in the sense that top-tier SOTA models (like the latest GPT, Claude, or Gemini series) can already achieve very high performance. This makes it difficult to assess how much of the performance is from the novel AutoRAS framework versus the underlying power of the base LLM it uses.
>
> **In both Table 1 and Table 2, we explicitly compare AutoRAS against the backbone model itself**. In Table 1 (GPT-4o-mini), AutoRAS improves over the backbone by **+16.27** under vanilla settings and **+21.76** under attack. In Table 2, across 4 different backbones, AutoRAS yields consistent improvements of **+9.40** (vanilla) and **+20.59** (attack) on MMLU. These results directly quantify the gains contributed by AutoRAS beyond the inherent capability of the underlying LLM and are not affected by potential benchmark saturation.
>
> ---
>
> > Weakness 2. Complexity of Optimization: The system uses flow-based optimization to search the sequence space, which can be computationally expensive and complex to train. The performance is also sensitive to parameters like sequence length and training samples per iteration.
>
> 1. **The optimization in AutoRAS is not computationally expensive in practice.** Our policy model has only **4.29M** parameters and occupies **49.22 MB** on disk, requiring roughly **450 MB** of GPU memory during training. Furthermore, as demonstrated in **Appendix H.6**, the training exhibits rapid and stable convergence, typically stabilizing after processing approximately 30 training trajectories. In our experiments, a full training run takes about **2–6 hours**, and more than **90%** of the total time is spent on LLM API calls.
> 2. **While primitive-sequence generation is indeed a non-trivial problem**, the formulation provides a sufficiently rich design space: with a maximum sequence length \(L=16\), it can represent approximately $10^6$ - $10^8$ distinct workflows by jointly modeling structural and behavioral patterns. At the same time, the effective search space is substantially reduced through primitive-level legality constraints based on stack rules, which help avoid invalid compositions and keep the search manageable and efficient (detailed in **sec. 4.2**).
> 3. **Our optimization design provides stable and sample-efficient learning in large discrete workflow spaces.** We adopt the **Trajectory Balance** (TB) objective, a single-step supervised loss that avoids long-horizon credit assignment and offers more stable training than prior work. Furthermore, **GFlowNets** optimize a distribution over valid workflows rather than forcing convergence to a single global optimum, which improves sample efficiency and stabilizes exploration in complex design spaces.
> 4. **Sensitive to parameters.** We have conducted a hyperparameter sensitivity analysis in **Sec. 5.5** on the MMLU dataset. To further clarify the effect of hyperparameters on our framework, we additionally performed the same analysis on **MSMARCO** and report the results in **Tables 1 and 2**. In fact, the hyperparameters used in our main experiments (L=16, K=4) are not necessarily optimal in terms of performance. However, across multiple benchmarks, we observe consistent and stable trends
> - When the maximum sequence length exceeds L = 16, further increases lead to only marginal improvements.
>
> - When the sampling count exceeds K = 4, performance does not increase meaningfully.
> These results demonstrate that L = 16 and K = 4 serve as practical and robust default settings. Overall, the results show stable hyperparameter sensitivity, which we further examine in **Appendix H.1**.
>
> **Tabel 1. Sensitivity to maximum sequence length \(L\).**
>
> | Dataset     |         | L=8   | L=12  | L=16  | L=20  | L=24  |
> | ----------- | ------- | ----- | ----- | ----- | ----- | ----- |
> | **MMLU**    | vanilla | 73.20 | 80.39 | 83.01 | 83.01 | 83.66 |
> |             | attack  | 72.55 | 75.82 | 82.35 | 81.70 | 81.70 |
> | **MSMARCO** | vanilla | 85.00 | 88.75 | 90.00 | 81.70 | 91.25 |
> |             | attack  | 82.50 | 88.75 | 88.75 | 87.50 | 88.75 |
>
> **Tabel 2. Sensitivity to sampling count \(K\).**
>
> | Dataset     |         | K=2   | K=4   | K=6   | K=8   |
> | ----------- | ------- | ----- | ----- | ----- | ----- |
> | **MMLU**    | vanilla | 81.17 | 83.01 | 83.01 | 83.01 |
> |             | attack  | 79.08 | 82.35 | 81.70 | 82.35 |
> | **MSMARCO** | vanilla | 85.00    | 90.00 | 91.25 | 91.25 |
> |             | attack  | 85.00    | 88.75 | 87.50 | 88.75 |

---

> ### Author Response · Authors · 2025-11-24
> **RESPONSE to Reviewer zZPz PART [2/3]**
>
> > Weakness 3. Missing Cost-Benefit Analysis: The AutoRAS algorithm is significantly more complex than a standard single-agent baseline. The paper does not provide a direct comparison of the computational cost against these simpler baselines, making it hard to evaluate if the performance and robustness gains justify the added complexity.
>
> We conducted a cost analysis on MMLU against 5 baselines (Table 3). AutoRAS can be viewed as **slightly more costly but highly effective**: it introduces a modest increase in training cost due to more fine-grained structural exploration and preventive control, yet it remains the **second-lowest** among all baselines, only behind MaAS. Most importantly, the robustness gain far outweighs this small overhead. AutoRAS achieves **83.01% → 82.35%** under attack, while other methods suffer much larger drops (**−14% to −27%**). This shows that the added structural search yields substantial robustness benefits relative to its cost.
>
> **Table 3. Cost analysis across baselines.** GPTSwarm, AgentPrune, and G-designer explicitly discuss robustness in their papers, while AFlow (ICLR 2025) and MaAS (ICML 2025) represent recent automated agentic system design methods.
>
> |  | Training  |  | | Inference  |  |   | Total| Performance ||
> |---|----|----|---|---|---|---|---|---| ------ |
> | **Method** | Prompt token | Completion token | Cost($) | Prompt token | Completion token | Cost($) | Cost($) | vanilla| attack |
> | GPTSwarm| 3,594,420| 1,065,580| 1.18 | 1,124,913| 430,268  | 0.43  | 1.61  | 75.82  | 71.24  |
> | Agentprune | 844,814 | 191,800  | 0.24  | 3,780,913 | 861,543  | 1.08  | 1.33  | 81.70 | 76.47  |
> | AFlow  | 10,117,493   | 1,153,666 | 2.21| 1,033,808| 161,119  | 0.25    | 2.46    | 82.35 | 70.58  |
> | G-designer | 598,010  | 115,704 | 0.15  | 2,840,066 | 528,981 | 0.74  | 0.90  | 82.35   | 73.86  |
> | MaAS | 392,428  | 306,049 | 0.24| 170,956  | 107,215 | 0.09   | 0.33    | 81.17  | 66.01  |
> | ours | 1,029,399 | 299,883| 0.33  | 978,251 | 307,540 | 0.33  | 0.67  | 83.01 | 82.35  |
>
> ---
> >
> > Weakness 4. Dependency on the Primitive Set: The entire framework's success is contingent on the quality and completeness of the predefined "primitive" vocabulary. If a highly effective or robust design pattern is not expressible using the existing primitives, AutoRAS will be unable to discover it.
>
> 1. The goal of AutoRAS is not to enumerate all possible agentic designs but to search for effective compositions within an expressive primitive space, following prior automated agentic system design work.  No existing representation, including graph, node, or operator, can theoretically cover all design patterns.  Our primitive sequence formulation already spans a large space: with L = 16, it can express roughly $10^6$~$10^8$distinct workflows by jointly modeling both structural and behavioral patterns.
>
> 2. GFlowNets naturally support equifinality, meaning that multiple workflow structures can achieve similar performance and reward.  Even if certain primitives are absent, AutoRAS can discover alternative patterns that achieve equivalent behavior.  Our experiments show that the current primitive set already enables state-of-the-art robustness and performance.  For instance, loop-like behaviors can be represented through unfolded sequential structures without requiring an explicit loop primitive.
> 3. As shown in Tabel 4, Primitive sequences provide a more flexible and expressive representation compared to prior representations.  The combination of structural primitives and behavioral primitives enables rich compositions while maintaining strong structural validity through stack rules.
> 4. The primitive set is extensible.  We conducted sensitivity analysis by varying the number and diversity of primitives, and the results (**Appendix D.3**) show that the current design is well-balanced: reducing primitives hurts performance, whereas adding many more yields only marginal improvement.  This indicates that the existing set is expressive and robust without being overly dependent on completeness.
>
> **Tabel 4. Comparison between primitives and other representation methods.**
>
> | Representation| Strengths| Limitations |
> |---| ---|---|
> | Neural Network   | Captures complex behavioral dependencies | Implicit structure and control flow  |
> | Graph | Clearly expresses workflow topology | Lacks behavioral semantics (e.g. reasoning mode or control conditions) |
> | Code| Highly expressive; precise control flow and arbitrary interaction logic | Fragile and difficult to constrain |
> | Operator | Clear behavioral semantics; easy to compose | No explicit structural semantics; Requires operator design   |
> | Primitive (Ours) | Unifies structural and behavioral semantics; compositional and searchable | Requires vocabulary design|

---

> ### Author Response · Authors · 2025-11-24
> **RESPONSE to Reviewer zZPz PART [3/3]**
>
> > Weakness 5. Dependency on LLMs: The framework's core optimization loop is critically dependent on the quality of its own internal LLMs. The "monitor" (Sec 4.2) uses an LLM to detect failures and generate safety signals; if this LLM fails to detect a novel or stealthy attack, the system cannot learn to defend against it. Similarly, the "analyzer" (Sec 4.1) and "textual gradient" (Sec 4.3) rely on an LLM to refine prompts, meaning the quality of the final agent behaviors is capped by this meta-LLM's capabilities.
>
> We agree that the performance of the monitor and analyzer matters, and our experiments show that the current design is effective. These components provide auxiliary signals rather than producing the workflow itself.
>
> **Monitor.** Detecting which component of an agentic system fails is a non-trivial problem, as emphasized by recent studies [1,2].
>
>  a). Our monitor follows the definitions and insights from this line of work and implements LLM-based detection of nine classes of internal failures (Sec. 4.2 and Appendix G). While such detection cannot be perfectly accurate due to the inherent difficulty, it has proven effective in practice.
>
> b). The monitor is also a plug-in module: it can be replaced with more specialized tools such as AgenTracer [3] once publicly available. To validate the effectiveness  of our method, we implemented alternative monitors from [1,2] (all-at-once, step-by-step, binary search, and MAST, details in **Appendix G**). As shown in Table 5, our monitor achieves the best balance of accuracy, robustness, and cost.
>
> **Table 5. Performance of different monitors on MMLU.**(all-at-once, step-by-step, and binary search from [1]; MAST from [2])
>
> | Monitor       | Vanilla | Attack | cost($) |
> | ------------- | ------- | ------ | ------- |
> | Ours          | 83.01   | 82.35  | 0.0815  |
> | all-at-once   | 80.39   | 75.16  | 0.0245  |
> | step-by-step  | 79.08   | 78.43  | 0.0623  |
> | binary search | 80.39   | 79.74  | 0.0496  |
> | MAST          | 83.01   | 79.74  | 0.3703  |
>
> **Analyzer.** The analyzer and the textual-gradient mechanism follow prior automated agent-design work such as MaAS (ICML 2025 Oral) and AFlow (ICLR 2025 Oral). We extend this design to incorporate additional safety and robustnes refinements. To evaluate the impact of different foundation models for the analyzer, we tested multiple alternatives (Table 6). Stronger analyzers produce only small gains but significantly increase cost, indicating that prompt refinement is not a difficult task and that the performance ceiling imposed by the analyzer is mild. Moreover, we argue that using a single foundation model for both the analyzer and the system reduces confounding effects introduced by heterogeneous models, ensuring that improvements stem from the framework.
>
> **Tabel 6.  Effect of analyzer backbone model.**
>
> | analyzer       | Vanilla | Attack | cost   |
> | -------------- | ------- | ------ | ------ |
> | GPT-4o-mini    | 83.01   | 82.35  | 0.0058 |
> | DeepSeek-V3.2-Exp   | 82.35   | 82.35  | 0.0077 |
> | GPT-5          | 84.97   | 84.31  | 2.5174 |
> | Gemini-2.5-pro | 83.66   | 83.01  | 0.4981 |
>
> [1] Which Agent Causes Task Failures and When?  On Automated Failure Attribution of LLM Multi-Agent Systems, ICML25.
>
> [2]Why Do Multi-Agent LLM Systems Fail?, NeurIPS25.
>
> [3]AgenTracer: Who Is Inducing Failure in the LLM Agentic Systems? ICLR 2026 under review.
>
> ---
>
> > Question. There are a number of typos such as line 047 and line 155.
>
> We have corrected the typos at the mentioned locations and updated the revised version accordingly.
>
> ---
>
> We sincerely hope that the clarifications and additional experiments provided above address your concerns!

---

### Author Response · Authors · 2025-12-02
**General Response Part[1/2]**

**Dear Area Chair,**

Thank you for the careful attention and substantial effort you have invested in managing the reviews, especially given the unexpected changes during the process. This comment summarizes the reviewers’ comments and outlines our point-by-point responses addressing all raised concerns. We hope this summary helps clarify the full review and rebuttal process.

### **High-Level Summary (for quick reading)**

Our paper proposes a principled framework (AutoRAS) built on a primitive-sequence representation that enables flexible structural topology and fine-grained behavioral choices for automatically designing robust agentic systems. Robustness is incorporated directly into the design loop through textual safety signals and flow-based optimization, moving beyond static strategy and post-hoc repair.

**Reviewers agree** that the formulation(zZPz, 2i3R), optimization method(zZPz, DBFH), and overall significance((zZPz,DBFH,2i3R,5XBL) are novel, sound, and relevant.  Remaining concerns largely involved **clarifications or additional analyses**, all of which we addressed with new experiments (cost, hyperparameter, primitive vocabulary, ablations, and production-grade comparisons).

We hope this note helps highlight the conceptual value and careful rebuttal work, and we are grateful for your consideration despite modest scores.

### **1. Our contribution provides a coherent and principled advancement in robust agentic-system design.**

The motivation behind our work comes from a clear and pressing challenge: **current agentic systems remain fragile because their structure and behavior are not designed in a unified and flexible way, and robustness is addressed only after failures occur.**
 This fragmentation makes it difficult to reason about agentic-system systematically, to anticipate adversarial or internal failures, or to explore the vast design space in a principled way.

To address this gap, **we introduce a unified, extensible abstraction that formulates agentic-system design as primitive-sequence generation**, directly aligning the method with the nature of the problem:

- **A unified primitive-based representation** consolidates topology, behavioral, and safety mechanisms into a single symbolic design space. This enables *joint reasoning* about structure and behavior—an ability that existing formulations lack.
- **A robustness-embedded design loop** incorporates execution-derived diagnostics into each successive design query, allowing the system to *actively learn* from vulnerabilities rather than reactively patch failures.
- **Flow-based trajectory-balance optimization** provides a principled way to explore a vast, highly non-convex design space. It naturally handles equifinality and avoids brittle, single-solution collapse by distributing probability mass across diverse yet high-quality workflows.

Together, **these components form a method tightly aligned with the core motivation**:
 to develop **a scalable, model-agnostic, and systematically robust framework for designing agentic systems**, validated consistently across tasks, backbones, and adversarial settings.

### **2. Reviewers share a clear consensus on the core strengths of the paper**

Across reviewers, there is **consistent acknowledgement of the conceptual clarity and methodological value** of our work.

* **Problem formulation** viewed as clear and meaningful(zZPz,2i3R)
* **Optimization framework** described as novel and sound(zZPz, DBFH, 2i3R)
* **Practical relevance** for agentic systems highlighted(DBFH, 5XBL)
* **Strong empirical robustness** observed(zZPz, 5XBL)
* **Cross-model and cross-dataset transferability** noted(zZPz)

While scores were modest, **the agreement on fundamental strengths suggests confidence in the core ideas**.

---

> ### Author Response · Authors · 2025-12-03
> **General Response Part[2/2]**
>
> ### **3. All reviewer concerns were clarificatory in nature and have been fully addressed**
>
> The raised issues did not challenge the correctness of the core method; instead, they sought clarification or additional evidence. We responded with new analyses and experiments, summarized below.
>
> - **Expressiveness and sensitivity of the Primitive Set**(2i3R-W2Q1, 5XBL-W2, zZPz-W4)
>
>   We clarified that AutoRAS aims to find strong designs within an *flexible*, *expressive* and *extensible* primitive space(rather than cover all possible workflows). Primitive sensitivity experiments indicates that the current primitive set is already robust and well-balanced.
>
> - **Optimization Clarification and Hyperparameter Sensitivity**(zZPz-W2, 5XBL-W3)
>
>   The policy model is lightweight (4.29M params, ~450 MB GPU) and exhibits rapid convergence, training is dominated by LLM latency rather than computation overhead. Sensitivity experiments on maximum sequence L and sampling times K across 3 benchmark confirm that our default settings (L=16, K=4) are stable, practical defaults rather than fragile.
>
> - **Cost analysis**(zZPz-W3, 2i3R-W1Q2)
>
>   Through a token-level cost analysis on MMLU, we demonstrated that AutoRAS has the second-lowest total cost, showing that the optimization is slightly more costly but clearly worthwhile.
>
> - **Flexibility of the Monitor and Analyzer)**(zZPz-W5)
>
>   We clarified that the monitor and analyzer serve as auxiliary signal generators, and both their backbone models and evaluation strategies are fully pluggable. Additional experiments comparing multiple analyzer backbones and monitor implementations show that the current configuration offers the best balance of effectiveness and cost.
>
> - **Comparison with Production-Grade Systems**(DBFH-W2)
>
>   AutoRAS follows the problem formulation in prior automated agentic system design work. Additionally, we also compared AutoRAS against 3 production-grade systems (Magentic-One, CAMEL, OWL), where AutoRAS achieved strong performance and the smallest drop under attack at competitive cost.
>
> - **Fairness of Robustness-Oriented Reward vs Baselines**(5XBL-W3)
>
>   We argued that using robustness signals in the reward is standard for agentic  system design, noted that several baselines also explicitly target robustness, and highlighted that AutoRAS differs mainly in performing dynamic, proactive robustness optimization rather than fixed, reactive heuristics.
>
> - **Risk of Overfitting**(5XBL-W3)
>
>   We clarified the theoretical basis and conducted additional experiments—including training-query sensitivity analysis and cross-dataset transfer evaluation, which show that AutoRAS learns a general workflow-design policy rather than memorizing a few specific designs.
>
> - **Key difference between prior works**(5XBL-W1)
>
>   We provide qualitative analyses across both the representation and the system-capability. This includes contrasting primitives with NN, graph, operator, and code, as well as differences in system capabilities such as topology search, behavioral variation, robustness design, and optimization strategies.
>
> Together, these additions reinforce that **the method is both conceptually sound and empirically well-supported**.
>
> ### **4. We respectfully hope the AC may consider the broader contribution of this work**
>
> We recognize that although reviewers did not assign high initial scores, their written assessments were generally positive regarding the core ideas, as summarized above.   These comments consistently acknowledged the clarity of our formulation, the novelty of the optimization design, and the practical relevance of embedding robustness into the system-construction process.
>
> Our core contribution lies in proposing a principled framework that treats agentic-system design as a joint optimization over structure and behavior, with robustness integrated directly into the design loop rather than applied post hoc.   We hope that this perspective, and the accompanying empirical validation can offer a useful foundation for future research on scalable, reliable agentic systems.
>
> We sincerely appreciate your time and consideration, and we hope our work may serve the community by helping shape more systematic approaches to agentic-system design.
>
> **Sincerely,
> The Authors**

---

### Meta-Review · Area_Chair_WYGq · 2025-12-24

**Summary:**

The summary of the reviewers' concerns that informed my suggested decision for this paper is shown as follows. 1)  The proposed method is overly complex. It combines a sequence generator, GFlowNets, Trajectory Balance, and a separate "analyzer" LLM for "textual gradients". It feels like a lot of moving parts just to generate a simple workflow; 2) The whole contribution seems to hinge on the "primitives"; 3)
Many of the components in the current production agents come from carefully assessing the performance of different part of the system and crafting solutions to improve the performance. A system such as AutoRAS cannot replicate this process.

**Reviewer Concerns:**

The concerns raised by Reviewer zZPz and Reviewer 2i3R are addressed by the authors, i.e., the authors perform related experiments for the Complexity of Optimization in the rebuttal and the Missing Cost-Benefit Analysis.

The concerns raised by Reviewer DBFH and Reviewer 5XBL are still outstanding, i.e.,  1)  The proposed method is overly complex. It combines a sequence generator, GFlowNets, Trajectory Balance, and a separate "analyzer" LLM for "textual gradients". It feels like a lot of moving parts just to generate a simple workflow; 2) The whole contribution seems to hinge on the "primitives"; 3)
Many of the components in the current production agents come from carefully assessing the performance of different part of the system and crafting solutions to improve the performance. A system such as AutoRAS cannot replicate this process.

**Reviewer Scores:**

The authors solve the concerns from Reviewer zZPz as above mentioned. Therefore, I think this reviewer tend to raise the score after the rebuttal. Likewise, the concerns from Reviewer 2i3R are also addressed and this reviewer tend to keep the original positive score.

Considering the concerns raised by Reviewer DBFH and Reviewer 5XBL are still outstanding as above pointed, I think these two reviewers tend to keep their original scores.

---

### Decision · Program_Chairs · 2026-01-26

Reject